# Cross-sectional study of prevalence and correlates of fear of falling in the older people in residential care in India: the Hyderabad Ocular Morbidity in Elderly Study (HOMES)

Srinivas Marmamula ,[1,2,3] Thirupathi Reddy Kumbham,[1] Satya Brahmanandam Modepalli,[1] Navya Rekha Barrenkala,[1] Jill Elizabeth Keeffe,[1] David S Friedman[4]

For numbered affiliations see end of article.

**Correspondence to**
Dr Srinivas Marmamula;
sri.marmamula@lvpei.org

## ABSTRACT

**Objective** To report the prevalence and risk factors for the fear of falling (FOF) among older individuals living in residential care facilities in India.

**Design** Cross-sectional study.

**Setting** Homes for the aged centres in Hyderabad, India.

**Participants** The study included individuals aged ≥60 years from homes for the aged centres. The participants underwent a comprehensive eye examination in make-shift clinics setup in homes. Trained investigators collected the personal and demographic information of the participants and administered the Patient Health Questionnaire-9 and Hearing Handicap Inventory for Elderly questionnaire in the vernacular language. FOF was assessed using the Short Falls Efficacy Scale. The presence of hearing and visual impairment in the same individual was considered dual sensory impairment (DSI). A multiple logistic regression analysis was done to assess the factors associated with FOF.

**Primary outcome measure** FOF.

**Results** In total, 867 participants were included from 41 homes for the aged centres in the analyses. The mean (±SD) age of the participants was 74.2 (±8.3) years (range 60–96 years). The prevalence of FOF was 56.1% (95% CI 52.7% to 59.4%; n=486). The multivariate analysis showed that those with DSI had eleven times higher odds of reporting FOF than those with no impairment (OR 11.14; 95% CI 3.15 to 41.4.) Similarly, those with moderate depression had seven times higher odds (OR 6.85; 95% CI 3.70 to 12.70), and those with severe depression had eight times higher odds (OR 8.13; 95% CI 3.50 to 18.90) of reporting FOF. A history of falls in the last year was also associated with increased odds for FOF (OR 1.52; 95% CI 1.03 to 2.26).

**Conclusion** FOF is common among older individuals in residential care in India. Depression, falling in the previous year and DSI were strongly associated with FOF. A cross-disciplinary approach may be required to address FOF among the older people in residential care in India.

## STRENGTHS AND LIMITATIONS OF THIS STUDY

⇒ Hyderabad Ocular Morbidity in Elderly Study (HOMES) is a cross-sectional study that included a large number of older people in residential care facilities in an urban region in India.

⇒ All participants underwent comprehensive assessments including visual acuity measurement, hearing assessment, depression, mobility and other assessments.

⇒ A multiple logistic regression analysis including visual impairment, hearing impairment and dual sensory impairment as covariates was done to assess the independent predictors of fear of falling in the older population.

⇒ The hearing assessment was based on a questionnaire, and this might have underestimated the prevalence of lower grades of hearing impairments.

⇒ The results of this study can only be generalised to the older population in residential care settings in urban regions in India.

## INTRODUCTION

The proportion of older people (people aged 60 years and older) in India's population is on the rise, and it is expected to reach 20% of the total population by 2050.[1] This demographic shift will dramatically change the nature of healthcare in India, and research on healthy ageing among the older population is needed. Such research can provide evidence for planning interventions to promote healthy and active ageing. Societal and lifestyle changes are sweeping through India, changing the living arrangements for the older population.[2 3] Multigenerational households are giving way to nuclear families and small households.[2 3] Under these changing circumstances, homes for the aged

or residential care facilities are emerging and taking root rapidly.[2 3]

Fear of falling (FOF) is a term used to describe an individual's loss of confidence in their ability to maintain balance, resulting in a worry about falling.[4] The prevalence of FOF in the older population ranges from 29% to 92% with a history of falls and 12%–65% without a history of falls.[5] In India, FOF is a common challenge affecting over one-third of the older population and affects their quality of life, physical well-being and social functioning.[6–8] Moreover, it is associated with depression,[9] cognitive decline and incidence of functional disability in the older population, it can also predict future falls.[10 11] Several risk factors are associated with FOF, including visual impairment (VI), hearing impairment (HI), a history of falls, depression and physical functioning.[12 13] A recent longitudinal study concluded that FOF is as detrimental as previous falls in limiting the daily activities among older people.[14]

Studies on active ageing, falls and FOF in older people are more common in high-income countries.[12 15] In India, such studies are still at a nascent stage, with very few studies reporting on FOF and none reporting on the association of FOF between dual sensory impairment (DSI, combined vision and hearing impairment) and depression among the older people in residential care.[6–8]

The Hyderabad Ocular Morbidity in Elderly Study (HOMES) is a large study conducted among older residents (aged ≥60 years) living in homes for the aged in Hyderabad, Telangana, India.[16] In addition to eye health examination, the study included an assessment of factors relevant to the health of older people, such as depression, falls, HI and FOF.[16] Previous publications from this study reported on distance and near VI, dual sensory loss, depression, prevalence, and risk factors for falls in older people.[17–19] In this paper, we report on the prevalence and associations of FOF with vision loss, hearing loss, dual sensory loss and depression among the older people living in homes for the aged facilities.

## MATERIAL AND METHODS
### Sampling method and recruitment of the participants
The FOF was assessed as a part of a larger eye health and VI.[16] A sample size of 916 participants was required based on an anticipated prevalence of 15% for avoidable VI, a 20% precision, a 25% non-response rate and a design effect of 1.4 to account for clustering for a cluster size of 40 people. Based on an anticipated prevalence of FOF of 33%, as reported by a previous study, the sample size required was 375 participants.[6] Of the 76 centres, 46 were selected (including five homes for the pilot) based on the proximity to a referral centre for eye care services and the willingness of the homes to participate in the study. Participants aged ≥60 years at the time of enumeration residing in the home for the aged for at least one month were included in the study.

### Eye examination and other assessments
The examination procedures have been described in previous publications.[17–20] Briefly, monocular visual acuity (VA) was assessed at a 3 m distance under ambient illumination using a logMAR ((logarithm of minimum angle of resolution) chart. The presenting and pinhole VAs were also assessed. Near vision was assessed using a logMAR chart at the standard distance of 40 cm. The anterior segment was examined using a portable slit lamp. A fundus examination and imaging were done for all participants and images were graded by trained graders. Mobility was classified as independently mobile and mobile with assistance based on the participants' self-report and the interviewers' observations, including using a walking stick or assisted by other people. The participants who were bedridden or in a wheelchair were not included in the study, though they were examined and provided with the necessary services.

### Non-clinical questionnaires
Trained investigators conducted detailed interviews with the participants using structured questionnaires.[16 20] All interviews were conducted before the eye examination. These included assessing the personal and demographic information (age, education level, marital status and type of home) and systemic history (diabetes and hypertension). The Hindi Mini-Mental State Examination (HMSE) was used to assess cognitive status.[21] All the participants with an HMSE score of 20 or more were further interviewed and assessed for depression using the Patient Health Questionnaire-9 (PHQ-9).[22] The hearing was assessed using the Hearing Handicap Inventory for Elderly (HHIE-S) questionnaire, with a hearing aid, if any.[23] Questionnaires were administered in the vernacular language (Telugu or Hindi).

FOF was assessed using the Short Falls Efficacy Scale (International) questionnaire.[24 25] This questionnaire has seven questions on commonly performed tasks, including self-care, physical activity and participation in social events. The response grades included not at all concerned, somewhat concerned, fairly concerned and very concerned. All the response grades were added to get a cumulative FOF score.[24] The history of previous falls was documented by asking, 'Have you ever fallen on the floor in the last 1 year?' in the vernacular language (Telugu or Hindi). The response was recorded as 0=no, 1=yes and 3=cannot remember.[18] The participants were also asked if they had a fall in the last two weeks.

### Definitions
1. FOF: A cumulative score ≥9 was considered as an individual with FOF. It was used as a dichotomous variable to define FOF. It was also graded as low (≤8), moderate (9–13) and high concern (14–28).
2. VI: Presenting distance VA worse than 6/18 in the better eye, without HI.
3. HI: It was defined as an HHIE score <10 on the HHIE-S, without VI.

4. DSI: The presence of both VI and HI in an individual was considered DSI.
5. Depression: An individual with a cumulative score of ≥10 on the PHQ-9 was considered as having depression. It was graded as none/mild (0–9), moderate (10–14) and moderate to severe/severe (14–17).
6. Multimorbidity: It was defined as reporting two or more non-communicable diseases in an individual.
7. Polypharmacy: It was defined as the use of five or more medications by an individual on a daily basis.

### Patient and public involvement
The patients and the public were not involved in the design and conduct of the study.

### Data management
Data analysis was conducted using Stata V.14.0 (StataCorp LP, College Station, Tx).[26] For descriptive statistics, continuous variables were analysed using the Student's t-test, and the categorical variables were analysed using the $\chi^2$ test. The prevalence of FOF was estimated and presented with 95% CIs. In the multiple logistic regression analysis, FOF was used as a dichotomous outcome variable, and its associations with personal and sociodemographic variables (age, gender and education), body mass index, multimorbidity, polypharmacy, depression, HI, VI and DSI were evaluated. The variables were entered into the model one at a time, and the selection of covariates for the model was based on previous studies reporting on FOF.[27] The Hosmer-Lemeshow goodness of fit test was used to assess the model fit. Variance inflation factors were used to test for collinearity between the covariates after fitting a multiple regression model. The adjusted ORs with 95% CIs were presented. Statistical significance was set at p<0.05 (two tailed), and the exact p-values were reported.

## RESULTS
### Characteristics of the study participants
The personal and demographic characteristics of the participants have been described in previous publications.[19] The HOMES study included 1182 participants; this included 271 participants who were bedridden and 98 participants with HMSE scores worse than 20.

Of the remaining 867 participants included in the analyses, 537 (61.6%) were women, 116 (13.4%) had no education, 518 (59.8%) reported hypertension and 263 (30.3%) reported diabetes. The mean (±SD) age of the participants was 74.2 (±8.3) years (range: 60–96 years). In terms of mobility status, 613 (70.7%) were independent, and the remaining 254 (29.3%) needed assistance from others or used walking aids for mobility. In total, 363 (41.9%) participants had multimorbidity and 161 (18.6%) reported polypharmacy.

### Sensory impairments
In total, 548 (63.2%) participants had no sensory loss, 134 (15.5%) had VI, 135 (15.6%) had HI and 50 (5.8%) had DSI. Among those with VI, the causes of VI were cataracts (44.0%; n=59), uncorrected refractive errors (35.5%; n=48), posterior capsular opacification (9%; n=12) and others (11%; n=15).

### Prevalence of FOF
The prevalence of FOF (moderate+high concern) was 56.1% (95% CI 52.7% to 59.4%; n=486). In total, 381 (43.9%) participants reported no concern about falling, 249 (28.7%) had a moderate concern and 237 (27.3%) reported a high concern. The prevalence of FOF varied with the nature of sensory loss, with the highest prevalence among those with DSI (92.0%), followed by HI (80.1%) and VI (57.5%). The prevalence of FOF among those with a history of falls was 69.4% (95% CI 63.2% to 75.2%) compared with 50.9% (95% CI 46.9% to 54.9%) among those without a history of falling. Those with multimorbidity had a significantly higher prevalence of FOF (table 1).

Figure 1 shows the grades of FOF among the participants without any impairment, HI, VI and DSI. The proportion of participants reporting a serious concern of falling was highest among those with DSI, followed by HI and VI (p<0.001).

### Risk factors for FOF
Table 2 shows the association between FOF and various risk factors. FOF was significantly associated with older age, high BMI and lack of education. The older participants with DSI had eleven times higher odds of FOF (OR 11.43; 95% CI 3.15 to 41.41, table 2) than those with no sensory impairment. HI was significantly associated with FOF (OR 3.34; 95% CI 1.96 to 5.72); however, VI (OR 1.51; 95% CI 0.95 to 2.40) was not significantly (p=0.08) associated. Those with moderate depression had seven times (OR 6.86; 95% CI 3.70 to 12.70) higher odds of FOF, and those with severe depression had eight times higher odds (OR 8.13; 95% CI 3.50 to 18.88). A history of falls over the past year, was also associated with an increase in odds for FOF (OR 1.53; 95% CI 1.03 to 2.26). Those needing assistance for mobility had five times higher odds of FOF (OR 4.84; 95% CI 3.1 to 7.44) compared with those with independent mobility. Polypharmacy and multimorbidity were not associated with FOF.

## DISCUSSION
Over half of the older people in residential care facilities had FOF. In addition, nearly 90% of those with combined HI and VI (DSI) reported FOF. Earlier studies done in India reported a lower prevalence of FOF compared with this study.[6–8] The possible reasons for a higher prevalence in this study could be explained by the differences in the living conditions of the older residents. Earlier studies reported on the older population living in a community setting, whereas those in this study live in an institutional setting. Reports from other countries have shown a higher prevalence of FOF among residents in institutions.[28] No

**Table 1** Personal, demographic and health-related characteristics and prevalence of fear of falling (n=867) among the older population in homes for the aged centres

| | Total in the sample | Fear of falling* n (row %) | Fear of falling — prevalence (95% CI) |
|---|---|---|---|
| Age group (years) | | | |
| 60–69 | 263 | 118 (44.9) | 9 (38.7 to 51.0) |
| 70–79 | 345 | 186 (53.9) | 53.9 (48.5 to 59.3) |
| 80 and above | 259 | 182 (70.3) | 70.3 (64.3 to 75.8) |
| Gender | | | |
| Male | 330 | 171 (51.8) | 51.8 (46.3 to 57.3) |
| Female | 537 | 315 (58.7) | 58.7 (54.4 to 62.9)) |
| Education level | | | |
| Any education | 751 | 400 (53.3) | 53.3 (49.6 to 56.9) |
| No education | 116 | 86 (74.1) | 74.1 (65.2 to 81.8) |
| Body mass index† | | | |
| Underweight/normal (≤24.9) | 397 | 210 (52.9) | 52.9 (47.8 to 57.9) |
| Overweight (25.0–29.9) | 289 | 159 (55.0) | 55.0 (49.1 to 60.8) |
| Obese (30 and above) | 153 | 97 (63.4) | 63.4 (55.2 to 71.0) |
| Hypertension | | | |
| No | 349 | 157 (45.0) | 45.0 (39.7 to 50.4) |
| Yes | 518 | 329 (63.5) | 63.5 (59.2 to 67.7) |
| Diabetes | | | |
| No | 604 | 345 (57.1) | 57.1 (53.1 to 61.1) |
| Yes | 263 | 141 (53.6) | 53.6 (47.4 to 59.7) |
| Mobility status | | | |
| Independently mobile | 613 | 275 (44.9) | 44.9 (40.9 to 49.9) |
| Mobility with support | 254 | 211 (83.1) | 83.1 (77.9 to 87.5) |
| Depression | | | |
| Mild/None | 672 | 313 (46.6) | 46.5 (42.7 to 50.4) |
| Moderate | 104 | 89 (85.6) | 85.6 (77.3 to 91.7) |
| Severe | 91 | 84 (92.3) | 92.3 (84.8 to 96.8) |
| History of fall in a year | | | |
| No fall | 625 | 318 (50.9) | 50.9 (46.9 to 54.9) |
| Falls reported | 242 | 168 (69.4) | 69.4 (63.2 to 75.2) |
| Vision impairment only | | | |
| No | 733 | 409 (55.8) | 55.8 (52.1 to 59.4) |
| Yes | 134 | 77 (57.5) | 57.4 (48.6 to 66.0) |
| Hearing impairment only | | | |
| No | 732 | 377 (51.5) | 51.5 (47.8 to 55.2) |
| Yes | 135 | 109 (80.1) | 80.1 (73.1 to 87.0) |
| Dual sensory impairment | | | |
| No | 817 | 440 (53.9) | 53.9 (50.4 to 57.3) |
| Yes | 50 | 46 (92.0) | 92.0 (80.8 to 97.8) |
| Multimorbidity | | | |
| No | 504 | 261 | 51.7 (47.3 to 56.2) |
| Yes | 363 | 225 | 62.0 (56.8 to 67.0) |
| Polypharmacy | | | |
| No | 706 | 382 | 54.1 (50.3 to 57.8) |
| Yes | 161 | 104 | 64.6 (56.7 to 72.0) |
| Total | 867 | 486 (56.1) | 56.1 (52.7 to 59.4) |

*Fear of falling was defined as a score of ≥9 on the Short Falls Efficacy Scale (International) questionnaire.
†BMI data were not available on 28 participants.
BMI, body mass index.

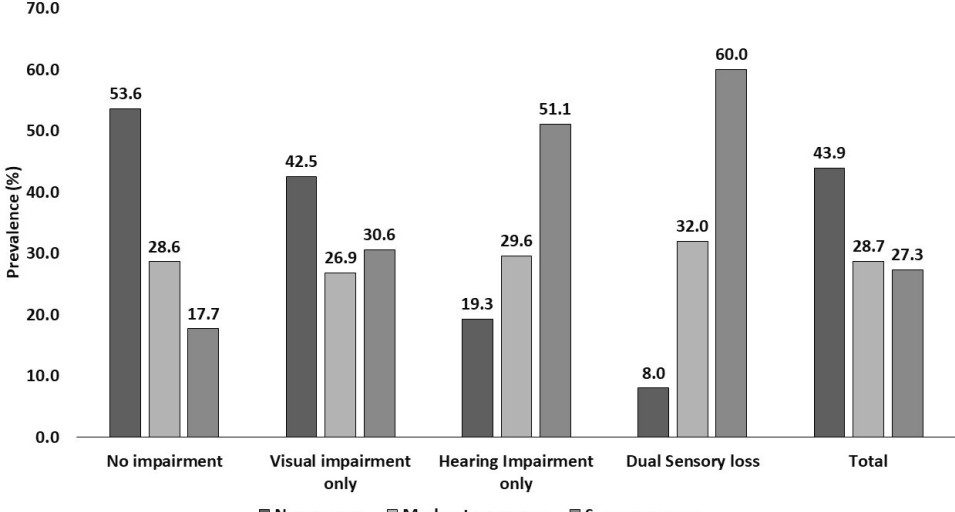

**Figure 1** Sensory loss and fear of falling among the older people in residential care in India.

previous studies have reported FOF among older residents in institutional care in India. One study reported a 42% prevalence of FOF among the older people in a hospital setting in India.[29]

Studies from other parts of the world have reported a higher prevalence of FOF among the older people living in institutional settings than those living in communities.[28 30 31] However, the results from studies done in high-income countries cannot be extrapolated to India. Nursing homes and homes for the aged in high-income countries are well established, with clear protocols in terms of infrastructure requirements, dedicated personnel specialised in care for older people and laws governing the quality of care. Homes for the aged in India are relatively new, and there are no standard minimum requirements for establishing such institutions.[32] The facilities, personnel and scope of service vary greatly based on the paying status of residents and the leadership of the facilities.[32] These homes are largely are run by private and non-government entities.[32] Due to these differences, the lifestyles of the older residents in homes in India and other high-income countries are not comparable.

The association between VI and FOF has been inconsistent across studies.[33–35] Some have reported a positive association[36] while others have found no associations.[35 37 38] There was no association between VI and FOF in this study after adjusting for other covariates. A possible reason for the inconsistent associations across the studies is the differences in the definition used for VI. However, there could also be a small association, which most studies are underpowered to detect. We did find a positive association, but it was not statistically significant. It is also possible that the studies that reported an association between VI and FOF could have included those with DSI, which may have increased the association, as people with DSI were more likely to report FOF. Those with DSI had a substantially higher FOF compared with those with single sensory loss, either VI or HI. A recent study on the

older population in nursing homes also reported a higher prevalence of FOF among those with DSI.[28]

The studies on eye health do not include the assessment of hearing as a routine procedure. We found that approximately 90% of those with DSI reported FOF, highlighting the need to assess both hearing and vision status in the older population for comprehensive care and cross referral. It is also useful to assess DSI when carrying out research studies in older individuals. Finally, though earlier studies reported a higher prevalence of FOF among women, there was no significant association with gender in this study.[39] FOF is associated independently with VI and HI.[34] A recent study reported a higher prevalence of FOF among nursing home residents with sensory loss, as over half of them had FOF.[28] A study by Murphy *et al* reported that older people who had restricted their daily activities out of FOF had a higher burden of depressive symptoms than those who only reported FOF.[40] A meta-analysis reported that untreated depression is significantly associated with an increased risk of falling, which in turn may lead to FOF.[27]

FOF was associated with depression in this study, which is consistent with other studies.[13 30 36 41–46] Causality is impossible to demonstrate using a cross sectional study design; however, several pathways for FOF leading to depression could be hypothesised. FOF is associated with reduced physical activity in older people.[47] This is evident in the current study where a higher odds for FOF among those needing assistance for mobility is noted. The reduced activity results in a more sedentary lifestyle, increasing social isolation and leading to depression. However, it is also possible that those with higher levels of depression tend to become sedentary and limit their social interaction and mobility.

In our previous publication, we reported that the odds of having depression is ten times higher in people with DSI than those with a single sensory loss.[19] In this study, HI was independently associated with FOF after adjusting

**Table 2** Multiple logistic regression analysis, assessing the factors associated with fear of falling in older people living in residential care (n=839)

|  | Adjusted OR (95% CI) | P value |
|---|---|---|
| Age group (years) |  |  |
| 60–69 | Reference |  |
| 70–79 | 1.27 (0.86 to 1.88) | 0.233 |
| 80 and above | 1.84 (1.16 to 2.92) | 0.010 |
| Gender |  |  |
| Male | Reference |  |
| Female | 1.35 (0.95 to 1.94) | 0.097 |
| Education |  |  |
| Any education | Reference |  |
| No education | 2.31 (1.34 to 3.99) | 0.003 |
| Body mass index |  |  |
| Normal | Reference |  |
| Overweight | 1.28 (0.87 to 1.87) | 0.215 |
| Obese | 2.03 (1.26 to 3.27) | 0.003 |
| Multimorbidity |  |  |
| No | Reference |  |
| Yes | 1.22 (0.85 to 1.78) | 0.273 |
| Polypharmacy |  |  |
| No | Reference |  |
| Yes | 1.36 (0.85 to 2.17) | 0.193 |
| Mobility status |  |  |
| Independent mobility | Reference |  |
| Mobility with support/aid | 4.84 (3.15 to 7.44) | <0.001 |
| Depression |  |  |
| None | Reference |  |
| Moderate | 6.86 (3.71 to 12.70) | <0.001 |
| Severe | 8.13 (3.50 to 18.88) | <0.001 |
| History of fall in last year |  |  |
| No | Reference |  |
| Yes | 1.53 (1.03 to 2.26) | 0.033 |
| Dual sensory impairment |  |  |
| No | Reference |  |
| Yes | 11.43 (3.15 to 41.42) | <0.001 |
| Vision impairment only |  |  |
| No | Reference |  |
| Yes | 1.51 (0.95 to 2.40) | 0.084 |
| Hearing impairment only |  |  |
| No | Reference |  |
| Yes | 3.35 (1.96 to 5.72) | <0.001 |

for other covariates. The possible explanation for this finding could be the changes in gait balance due to HI, leading to an increased FOF. A greater FOF was reported among those with lower gait velocity.[48]

Independent mobility is a measure of physical function, and a loss of it is associated with falls and FOF.[49–51] Poor physical function may result in a loss of confidence among older people, resulting in an increased FOF.[50] In a longitudinal study, FOF was associated with a lack of independent mobility and poor physical performance at the end of two years.[14] This indicates the need to address FOF in the older population to prevent a decline in physical functions.

The association between falls and FOF is reported in a few studies in India.[7 37] FOF was higher among those with a history of falls in India.[7 33] Adverse consequences of falls in the older people are also well known, which in itself may lead to FOF. Fear is an innate response that has an evolutionary importance and is a protective instinct. Similarly, FOF could act as an instinctual defence mechanism to prevent falls but only to a certain threshold.[31 52] Beyond this threshold, FOF could impair physical functioning and negatively impact the quality of life in older people.[31 52] A recent systematic review has concluded that FOF adversely impacts the quality of life of older residents in community settings and institutional care.[53]

Several interventions have been reported to address FOF in older people living in communities and nursing homes. These interventions include cognitive and behavioural interventions, occupational therapy and physical exercises, including balance training, yoga and tai chi.[9 43 54–59] In addition to addressing FOF, these interventions may also have a resonating effect, alleviating depression, leading to better quality of life and promoting well-being in the older population.[60–62] However, studies on the impact of such interventions that address FOF are limited in India. The older residents living in institutional care present a unique opportunity where interventions can be studied systematically and applied to more individuals in one setting, compared to older population diffused across communities. Studies also indicate an association between FOF and cognitive decline,[36 63] which underpins the need to address FOF among the older people.

This study benefits from a large cohort of older participants and a comprehensive assessment (inclusion of DSI and depression). Some limitations include our decision to exclude participants with severe mobility issues and those with cognitive impairment, as indicated by the lower HSME scores. This may have led to an underestimation of the prevalence of FOF in our study. However, it is unlikely to have led to an inaccurate assessment of associations. Environmental and infrastructure factors also influence FOF, which were not evaluated in our study. A direct extrapolation of our results to those who live in the communities is limited due to the differences in the types of residences and the activities of people. Moreover, while vision assessment was done using the standard procedures used in the clinic, hearing was assessed using a questionnaire, which is not an objective assessment. This limitation could have led to an underestimation of the prevalence of HI in our study. HOMES was an observational study hence causality cannot be established.

In conclusion, FOF is common among the older population in residential care in India. It is especially common among those with depression and DSI. With the increase

in the older population and the rapid increase in the number of homes for the aged in urban areas, these findings are relevant for policy and healthcare service planning for older people. The findings of this study can help inform planners to develop appropriate interventions to address FOF in the older population and promote healthy and active ageing among these residents.

**Author affiliations**
[1]Allen Foster Community Eye Health Research Centre, Gullapalli Pratibha Rao International Centre for Advancement of Rural Eye care, L V Prasad Eye Institute, Hyderabad, Telangana, India
[2]School of Optometry and Vision Science, University of New South Wales, Sydney, New South Wales, Australia
[3]Wellcome Trust / Department of Biotechnology India Alliance, L V Prasad Eye Institute, Hyderabad, Telangana, India
[4]Harvard Medical School Department of Ophthalmology, Massachusetts Eye and Ear, Boston, MA, USA

**Acknowledgements** The authors thank the individuals for their participation in the study. Mr Shashank Yellapragada, Rajesh Challa and Madhuri Bakki are acknowledged for their assistance in data collection. Ratnakar Yellapragada and Ms. Muni Rajya Lakshmi are acknowledged for her support with data management. The authors also thank Mr Abhinav Sekar for his language inputs on earlier versions of the manuscript.

**Contributors** SM conceived the idea, designed and conducted the study, analysed the data and wrote the manuscript. TRK, SBM and NRB were involved in data collection and data management and provided intellectual inputs on the earlier versions of the manuscript. JEK and DF critically reviewed the manuscript and provided intellectual inputs. SM bore primary responsibility for the final content of the manuscript and the accuracy of the data. SM, TRK, SBM, NRB, JEK and DSF approved the final manuscript. SM is responsible for the overall content as a guarantor.

**Funding** This work was supported by Wellcome Trust/DBT India Alliance Fellowship (IA/CPHE/14/1/501506) awarded to SM and Hyderabad Eye Research Foundation (HERF), India.

**Competing interests** None declared.

**Patient and public involvement** Patients and/or the public were not involved in the design, or conduct, or reporting, or dissemination plans of this research.

**Patient consent for publication** Not applicable.

**Ethics approval** The protocol for the HOMES study was approved by the Institutional Review Board of the Hyderabad Eye Research Foundation, L V Prasad Eye Institute, Hyderabad, India (reference number: LEC-08-16-073). The study was performed in accordance with the Declaration of Helsinki. Each participant provided written informed consent. The data collection was carried out during 2017–2019.

**Provenance and peer review** Not commissioned; externally peer reviewed.

**Data availability statement** No data are available.

**ORCID iD**
Srinivas Marmamula http://orcid.org/0000-0003-1716-9809

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
