## [Reviewer comments · BMJ Open]

ARTICLE DETAILS

TITLE (PROVISIONAL)	A cross sectional study of prevalence and correlates of fear of falling in the older people in residential care in India – the Hyderabad Ocular Morbidity in Elderly Study
AUTHORS	Marmamula, Srinivas; Kumbham, Thirupathi Reddy; Modepalli, Satya Brahmanandam; Barrenkala, Navya Rekha; Keeffe, Jill; Friedman, David

VERSION 1 – REVIEW

REVIEWER	Menant, Jasmine University of New South Wales, School of Public Health and Community Medicine
REVIEW RETURNED	08-Nov-2023

GENERAL COMMENTS	This paper reports on the results of a cross-sectional analysis on the prevalence and factors associated with fear of falling among 867 older people aged 60 year and over living in aged care residencies in India. The paper is overall nicely written. The study includes a large sample of older aged care residents. The sampling is well defined. However, i am a bit unclear about the analysis. Please find below my specific comments below. Methods Page 5. The eligibility criteria for participation in this study should be clearly stated. Page 6. Please state what does a cut-point score of 20 on the Hindu MMSE indicate. It is not clear why only the participants with a cut-point of 20 on the MMSE were further assessed with the PHQ-9. Please clarify how many participants had a Hindu MMSE score <20 and if they were included in the multiple logistic regressions. Please specify what covariates were entered in the multiple logistic regressions. Results The term “more likely” is erroneously used in several places in the abstract and results section when reporting the results of an association using odds ratios. The term “increased / decreased odds” should be used. The terms “more likely” is used together with relative risks or likelihood ratios. Table 1: The prevalence of fear of falling appears significantly higher in those classified as “independently mobile” (83.1%) as opposed to “mobile with support” (44.9%). However the odds ratio reported in Table 2 appears to indicate that those with a mobility aid had significantly greater odds of experiencing fear of falling (OR=4.84). Please clarify. Are the data presented in Table 2 the results of multiple multivariate models or univariate models? It is a little unclear. If
---

	univariate models, could the authors then conduct one or several multivariate regression models to determine the independent predictors of fear of falling in this population, using the factors that came out as significant in the univariate associations? This would make the results and conclusions more robust.
REVIEWER	OGLIARI, GIULIA Nottingham University Hospitals NHS Trust, Health Care for Older People (HCOP)
REVIEW RETURNED	22-Jan-2024
GENERAL COMMENTS	The paper by Marmamula explores the cross-sectional associations between fear of falling (FOF) and dual sensory impairment and depression among 867 older adults in residential care facilities in India. My major comment is to delete the word “elderly” throughout the text and substitute it with “older people” or “older participants”. Discussion, page 14. The term “mushrooming” is a bit too informal. Please, consider changing it. Discussion. in the section on the limitations of the study, the Authors should acknowledge that their study is observational. Their study can "inform", rather than "help plan and execute appropriate interventions to address FOF".

VERSION 1 – AUTHOR RESPONSE

Reviewer: 1

Dr. Jasmine Menant, University of New South Wales

Comments to the Author:

This paper reports on the results of a cross-sectional analysis on the prevalence and factors associated with fear of falling among 867 older people aged 60 year and over living in aged care residencies in India. The paper is overall nicely written. The study includes a large sample of older aged care residents. The sampling is well defined. However, I am a bit unclear about the analysis. Please find below my specific comments below.

Response: Thank you for your comments.

Methods

Page 5. The eligibility criteria for participation in this study should be clearly stated.

Response: Thank you for highlighting this. We have provided the eligibility as follows: Participants aged ≥ 60 years at the time of enumeration residing in the home for the aged for at least one month were included in the study.

Page 6. Please state what does a cut-point score of 20 on the Hindu MMSE indicate. It is not clear why only the participants with a cut-point of 20 on the MMSE were further assessed with the PHQ-9. Please clarify how many participants had a Hindu MMSE score < 20 and if they were included in the multiple logistic regressions.

Response: Thank you for this important comment. Lower Hindi HMSE scores are suggestive of cognitive impairment. Hence, we did not conduct the PHQ9 on the participants with lower scores. The HOMES study included 1182 participants; this included 271 participants who were bedridden and 98 participants with HMSE scores less than 20. The data of the remaining 867 participants was used for this manuscript.

Of 867 people, data on BMI was missing on 28 people and regression analysis included 839 participants. To clarify this point, we have the sample size in the table heading in our revision.

Please specify what covariates were entered in the multiple logistic regressions.

Response: In the multiple logistic regression analysis, FOF was used as a dichotomous outcome variable; The covariates entered in the model included age group, gender, level of education, Body Mass index, multimorbidity, polypharmacy, mobility status, depression, history of fall in last one year, dual sensory impairment, vision impairment only and hearing impairment only.

Results

The term "more likely" is erroneously used in several places in the abstract and results section when reporting the results of an association using odds ratios. The term "increased / decreased odds" should be used. The terms "more likely" is used together with relative risks or likelihood ratios.

Response: Thank you for this comment. We have modified the results of our regression as per suggested.

Table 1: The prevalence of fear of falling appears significantly higher in those classified as "independently mobile" (83.1%) as opposed to "mobile with support" (44.9%). However the odds ratio reported in Table 2 appears to indicate that those with a mobility aid had significantly greater odds of experiencing fear of falling (OR=4.84). Please clarify.

Response: Thank you for spotting this difference which allowed us to review our data and correct it. There was an error in labelling in Table 1 which is corrected in revision. The correction prevalence is 83.1% among those who were mobile with support and 44.9% among those who were independent.

Are the data presented in Table 2 the results of multiple multivariate models or univariate models? It is a little unclear. If univariate models, could the authors then conduct one or several multivariate regression models to determine the independent predictors of fear of falling in this population, using the factors that came out as significant in the univariate associations? This would make the results and conclusions more robust.

Response: Thank you for this valuable comment. In Table 2, we have presented the results from the multivariate model conducted to understand the independent predictors for fear of falling in our study population.

Thank you for the valuable comments to improve our manuscript.

Reviewer: 2

Dr. GIULIA OGLIARI, Nottingham University Hospitals NHS Trust

Comments to the Author:

The paper by Marmamula explores the cross-sectional associations between fear of falling (FOF) and dual sensory impairment and depression among 867 older adults in residential care facilities in India.

My major comment is to delete the word "elderly" throughout the text and substitute it with "older people" or "older participants".

Response: As suggested, we have changed the term 'elderly' to older people or older participants throughout the manuscript.

Discussion, page 14. The term "mushrooming" is a bit too informal. Please, consider changing it.

Response: Thank you for this suggestion. We rephrase the statement deleting the term 'mushrooming'. The revised statement reads as follows:

With the increase in the older population and the rapid increase in the number of homes for the aged in urban areas, these findings are relevant for policy and healthcare service planning for older people.

Discussion. In the section on the limitations of the study, the Authors should acknowledge that their study is observational. Their study can "inform", rather than "help plan and execute appropriate interventions to address FOF".

Response: Thank you for the comments. We have acknowledged the observational nature of our study in our revision. We have rephrased the statement in the conclusion as suggested. Thank you for the valuable insights that have improved our manuscript.